# Selective photocatalytic conversion of methane into carbon monoxide over zinc-heteropolyacid-titania nanocomposites

Xiang Yu [1], Vincent De Waele[2], Axel Löfberg[1], Vitaly Ordomsky[1,3] & Andrei Y. Khodakov [1]

Chemical utilization of vast fossil and renewable feedstocks of methane remains one of the most important challenges of modern chemistry. Herein, we report direct and selective methane photocatalytic oxidation at ambient conditions into carbon monoxide, which is an important chemical intermediate and a platform molecule. The composite catalysts on the basis of zinc, tungstophosphoric acid and titania exhibit exceptional performance in this reaction, high carbon monoxide selectivity and quantum efficiency of 7.1% at 362 nm. In-situ Fourier transform infrared and X-ray photoelectron spectroscopy suggest that the catalytic performance can be attributed to zinc species highly dispersed on tungstophosphoric acid / titania, which undergo reduction and oxidation cycles during the reaction according to the Mars–van Krevelen sequence. The reaction proceeds via intermediate formation of surface methyl carbonates.

[1] Université Lille, CNRS, Centrale Lille, ENSCL, Université Artois, UMR 8181 - UCCS - Unité de Catalyse et Chimie du Solide, 59000 Lille, France. [2] Université Lille, CNRS, UMR 8516, LASIR, Laboratoire de Spectrochimie Infrarouge et Raman, 59000 Lille, France. [3] Eco-Efficient Products and Processes Laboratory (E2P2L), UMI 3464, CNRS-Solvay, 201108 Shanghai, People's Republic of China. Correspondence and requests for materials should be addressed to V.O. (email: vitaly.ordomsky@univ-lille.fr) or to A.Y.K. (email: andrei.khodakov@univ-lille.fr)

In recent years, methane has become increasingly abundant due to the development of shale gas fields and other cost efficient or renewable feedstocks such as biogas. Methane is also considered as one of the greenhouse gases with a global warming potential 50 times higher than carbon dioxide[1]. Methane activation is therefore a formidable challenge for catalysis[2–5]. High reaction temperatures (>700 °C), low selectivity to target products, and often abundant $CO_2$ production are major drawbacks of conventional thermocatalytic technologies.

Photocatalysis, which uses sunlight, has been shown to be very promising for water decomposition and environmental remediation. Photocatalysis has been also considered as one of pathways to break the thermodynamic barrier[6–15]. Only a few examples of methane photocatalytic conversion are available in the literature. Earlier reports have shown that methane can be converted by photocatalytic steam reforming to hydrogen ($CH_4 + 2H_2O \rightarrow 4H_2 + CO_2$)[8–13] or can undergo photocatalytic total oxidation ($CH_4 + 2O_2 \rightarrow CO_2 + 2H_2O$)[9]. A limited number of papers[16,17] have also addressed combined photo-thermocatalytic[18] or plasma-enhanced[19] methane dry reforming, which represents an interesting route for production of carbon monoxide and hydrogen. A few reports[20–23] also suggest that methane photo-oxidation can produce methanol, though extremely low yields have been achieved.

Carbon monoxide is a very important compound and a building block in chemical industry. It is utilized as a feedstock in the production of chemicals ranging from acetic acid to polycarbonates and polyurethanes. Syngas, which is a mixture of carbon monoxide and hydrogen, is a valuable feedstock for manufacturing methanol, hydrocarbon fuels, oxo-alcohols and aldehydes. CO is also an important reducing agent and it is used for manufacturing pure metals and in particular iron, cobalt and nickel.

Herein, we report direct selective photocatalytic conversion of methane into carbon monoxide under ambient conditions with only marginal $CO_2$ production:

$$2CH_4 + 3O_2 \xrightarrow{h\nu} 2CO + 4H_2O \qquad (1)$$

$$2CH_4 + O_2 \xrightarrow{h\nu} 2CO + 4H_2 \qquad (2)$$

A series of catalysts are developed on the basis of metals, $H_3PW_{12}O_{40}$ heteropolyacids (HPW) and $TiO_2$ (P25). The Zn-HPW/$TiO_2$ system exhibits exceptional photocatalytic activity in selective carbon monoxide production from methane. Zinc

species seem to play an important role in methane activation[6,24]. Importantly, methane activation and reaction are carried out at ambient temperature. High carbon monoxide yields (up to 3–4% in a single batch experiment), high quantum efficiency (QE = 7.1% at 362 nm) and extended catalyst stability make it potentially interesting in the future for practical applications. To the best of our knowledge, the present work presents the first example of utilizing photocatalysis for methane selective oxidation into carbon monoxide. The conducted in-situ investigation of the reaction mechanism is indicative of zinc reduction by methane with important modifications of the catalyst structure. Exposure to oxygen leads to subsequent regeneration of the composite catalyst according to Mars–van Krevelen mechanism[25–27].

## Results

### Catalytic performance of the metal HPW/$TiO_2$ composites.
Photocatalytic oxidation of methane was investigated on $TiO_2$, HPW, HPW/$TiO_2$ composites containing different metals (Fig. 1a). Two products, CO and $CO_2$, were detected on these solids after light irradiation using a 400 W Xe lamp. Trace amounts of hydrogen were also observed, while no methanol was detected.

Addition of noble or transition metals (Ag, Pd, V, Fe, Ga, Ce, Co, Cu and Zn) to HPW/$TiO_2$ strongly affects the rate and selectivity of methane oxidation. Much higher activity was observed over the catalysts containing noble metals (Fig. 1a), however, this higher activity was accompanied by significant carbon dioxide production. Note that only $CO_2$ was detected in methane photo-oxidation over the Pd containing catalyst. The silver containing HPW/$TiO_2$ composite also exhibited extremely high activity in methane photo-oxidation with carbon dioxide as a major product. Among the transition metals, higher CO selectivity was observed on the zinc and copper based catalysts. Interestingly, most of metal HPW/$TiO_2$ composite catalysts also exhibited high activity in carbon monoxide oxidation to $CO_2$, except for the Zn and Cu catalysts (Supplementary Fig. 1b).

Compared to the Cu-based catalyst, Zn-HPW/$TiO_2$ in addition to high CO selectivity, demonstrated a very high activity in methane photocatalytic oxidation. Importantly, addition of zinc specifically promotes the carbon monoxide formation rate, which increased almost twenty times from 0.02 mmol g$^{-1}$ h$^{-1}$ over the pristine HPW/$TiO_2$ composite to 0.429 mmol g$^{-1}$ h$^{-1}$ over the catalysts doped with Zn. Remarkably, the CO selectivity reaches more than 84% on the Zn-HPW/$TiO_2$ catalyst. Clearly, among the studied catalytic systems, the Zn-HPW/$TiO_2$ catalyst has the

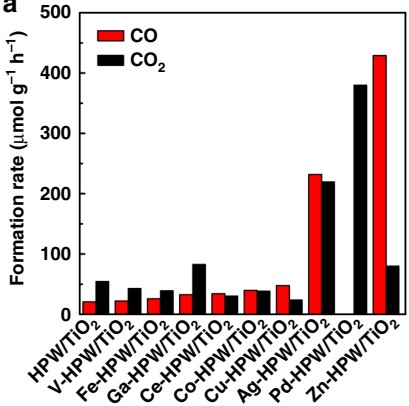

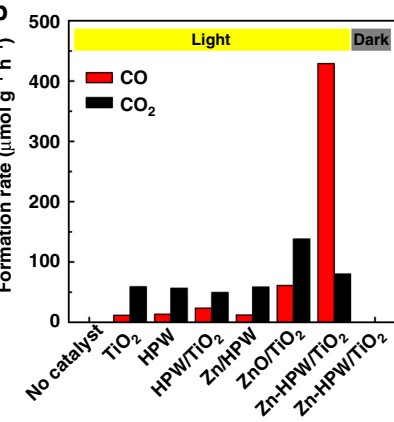

**Fig. 1** Methane photocatalytic oxidation on different catalysts. **a** Metal-HPW/$TiO_2$ composite, **b** $TiO_2$, HPW, and Zn containing catalysts. Reaction conditions: catalyst, 0.1 g; gas phase pressure, $CH_4$ 0.3 MPa, Air 0.1 MPa; irradiation time, 6 h

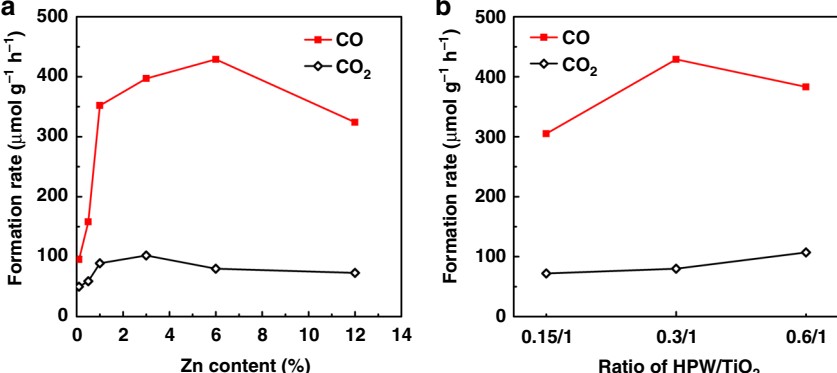

**Fig. 2** Carbon monoxide and carbon dioxide production over Zn-HPW/TiO$_2$ catalysts. **a** Influence of different Zn loadings, **b** influence of HPW/TiO$_2$ ratio. Reaction conditions: catalyst, 0.1 g; gas phase pressure, CH$_4$ 0.3 MPa, Air 0.1 MPa; irradiation time, 6 h

highest potential for methane selective photocatalytic oxidation to CO. The experiments without irradiation (dark) provided no products, confirming that the CO and CO$_2$ productions are indeed driven by light over the Zn-HPW/TiO$_2$ catalysts. In order to evaluate the influence of UV, visible and IR light on the methane conversion, we conducted photocatalytic experiments on selected spectral ranges (280 < $\lambda$ < 400 nm and $\lambda$ > 380, Supplementary Table 1). The results show that the catalyst is very sensitive to the irradiance spectral range. The catalyst provides only very mild activity under visible irradiation, while the reaction rate increases 20 times, when the reactor is exposed to UV.

Methane partial oxidation to CO can be compared with methane dry reforming. Methane dry reforming usually involves both thermo- and photo-catalysis. Note that in our work, methane photooxidation to carbon monoxide occurs with high selectivity at ambient temperature, while in previous reports[16,17], methane dry reforming was conducted at relatively high temperatures in order to obtain noticeable conversion.

The exceptional photocatalytic activity of the Zn-HPW/TiO$_2$ catalyst seems to be related to zinc species. It is reasonable to suggest that the overall reaction rate on Zn-HPW/TiO$_2$ can be affected by zinc content. Figure 2a illustrates the catalytic behavior for methane photocatalytic conversion of Zn-HPW/TiO$_2$ with different Zn loadings. Addition of even small amounts of zinc results in a major increase in the rate of methane conversion. Note that the presence of zinc principally increases the rate of methane conversion to CO, while the rate of methane oxidation to CO$_2$ is only slightly affected. Thus, formation of CO$_2$ might be explained by the activity of HPW/TiO$_2$, while the Zn species seem to be active and selective in methane photo-oxidation to CO. Note that the major increase in CO is only observed when the Zn content is higher than 2–3 wt.%. This amount corresponds to the amount of zinc, which can neutralize the acid sites in HPW. The highest rate of methane oxidation was observed at Zn content of 6.0 wt.%.

Another catalyst parameter, which may affect the catalytic performance is the HPW/TiO$_2$ ratio in the composite. Figure 2b shows the performance of Zn-HPW/TiO$_2$ catalysts with different ratio of HPW to TiO$_2$, while the molar ratio of Zn to HPW was kept at 2. The HPW/TiO$_2$ ratio in the composite Zn-HPW/TiO$_2$ catalysts does not noticeably affect the rate of CO$_2$ formation, while the effect of this ratio on the rate of CO formation is more significant. Note that only the CO production rate is strongly influenced by the concentration of highly dispersed Zn species. It is expected that higher HPW/TiO$_2$ ratio could enhance zinc dispersion, because of possible localization of zinc ions in the cationic sites of HPW. Some small decrease in the rate of CO

production at higher HPW/TiO$_2$ ratio can be due to the formation of larger HPW clusters, which would affect the electron transfer from TiO$_2$ to the Zn species.

Further, photocatalytic methane oxidation was investigated on the HPW and TiO$_2$ nanocomposites with and without zinc (Supplementary Fig. 2). TiO$_2$, HPW, and HPW/TiO$_2$ exhibit 10–20 times lower activity compared to Zn-HPW/TiO$_2$. The selectivity pattern was also very different. The methane photo-oxidation on TiO$_2$, HPW, and HPW/TiO$_2$ primarily results in CO$_2$, while CO was the major product over Zn-HPW/TiO$_2$. This suggests different mechanism and kinetics of methane photo-catalytic oxidation. The lattice oxygen activated by photo-generated hole could be the main active species for the activation of methane and oxygen and subsequent oxidation of the CH$_3$ radicals to CO$_2$ over the semiconductors without zinc[9].

Promotion of pure TiO$_2$ or HPW with Zn results only in a slight increase in the methane oxidation rate compared to the pristine semiconductors, whereas CO$_2$ remains the major reaction product. The mediocre catalytic performance of Zn/TiO$_2$ and Zn/HPW can be due to the following phenomena. First, Zn/TiO$_2$ can contain relatively large ZnO crystallites. Because of poor zinc dispersion, the uncovered TiO$_2$ surface leads to an important contribution of TiO$_2$ to methane total oxidation to CO$_2$. Second, in the absence of TiO$_2$, the reaction rate is low on Zn-HPW, principally because of low light harvesting. TiO$_2$ is a semiconductor, which is very efficient in light harvesting and charge separation. HPW could be efficient in transfer of holes and electrons from TiO$_2$ to Zn sites[28].

Thus, a major increase in CO production from methane only occurred when the composite catalyst contained together three components: TiO$_2$, HPW, and Zn. The observed strong effect of Zn on the catalytic performance might be therefore due to the intimate contact between Zn, HPW and TiO$_2$. It is expected that the interaction of ZnO with the H$_3$PW$_{12}$O$_{40}$ acid results in formation of Zn$^{2+}$ ions and possibly small positively charged Zn cationic nanoclusters. The highly dispersed Zn species may have high mobility in the composites. In order to confirm this, we have prepared mechanical mixtures of Zn/TiO$_2$ with HPW/TiO$_2$ and performed photocatalytic conversion of methane. The activity of these mechanical mixtures (Supplementary Fig. 2) was much higher than over either Zn/TiO$_2$ or HPW/TiO$_2$. This could indicate substantial migration of zinc during the reaction and formation of the active sites with the enhanced performance of methane oxidation to CO.

**Characterization of the Zn-HPW/TiO$_2$ catalysts.** The Zn-HPW/TiO$_2$ samples have shown the best catalytic performance in

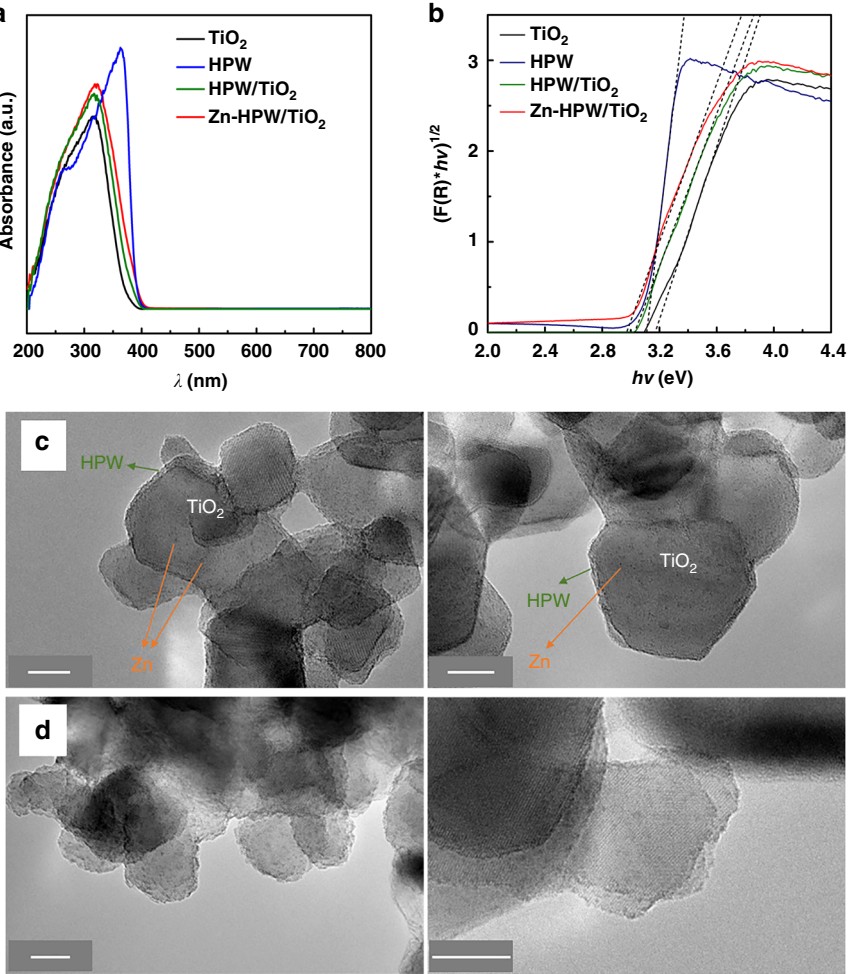

**Fig. 3** UV-visible spectra and TEM images. **a** UV-Vis spectra and **b** [F(R∞)$h\nu$]$^{1/2}$ versus $h\nu$ for Zn-HPW/TiO$_2$ and reference compounds; TEM images of calcined (**c**) and used (**d**) Zn-HPW/TiO$_2$. Scale bar: 10 nm

partial methane photocatalytic oxidation to carbon monoxide. A combination of techniques was used for their characterization. The X-ray diffraction (XRD) patterns of Zn-HPW/TiO$_2$, ZnO/TiO$_2$, HPW/TiO$_2$, TiO$_2$, and HPW are shown in Supplementary Fig. 3. The samples containing TiO$_2$ exhibit XRD peaks of anatase and rutile phases, while the XRD peaks assigned to the heteropolyacid are present in HPW. Interestingly, no diffraction peaks were detected for the HPW and Zn phases in Zn-HPW/TiO$_2$. This can be probably due to their high dispersion and smaller crystallite sizes. The XRD patterns are slightly different for the ZnO/TiO$_2$ sample. They show the presence of the hexagonal wurtzite ZnO phase (orange bar, JCPDS #36–1451), which was identified by diffraction peaks at 31.8° and 34.4° attributed to crystal face (100) and (002), respectively[9,29]. Note that the XRD peaks of ZnO almost disappear for the Zn-HPW/TiO$_2$ catalysts. This confirms that in the presence of HPW, zinc species are highly dispersed. No XRD peaks attributed to zinc carbonate were observed for any Zn-containing catalyst.

FTIR analysis has been used to identify the acidity and state of Zn in the catalyst. Supplementary Fig. 4 shows FTIR spectra of the initial Zn-HPW/TiO$_2$ catalyst after evacuation at 200 °C. The catalyst shows strong bands at 1560 and 1285 cm$^{-1}$, which might be assigned to $\nu_{as}(CO_3)$ and $\nu_s(CO_3)$ of bidentate carbonate species of Zn[30].

Adsorption of pyridine (Py) results in appearance of strong bands at 1621 and 1453 cm$^{-1}$, which might be attributed to the Py adsorption over strong Lewis acid sites[31]. The Lewis acid sites

were attributed to unsaturated Zn$^{2+}$ ions. No Brönsted acidity associated to HPW was observed in the Zn-HPW/TiO$_2$ sample. This catalyst contained 0.9 mmol of Zn, while the maximum concentration of potential Brönsted acid sites associated with HPW can be only 0.3 mmol g$^{-1}$ (assuming that the Brönsted acid sites are not neutralized by zinc). Thus, in the case of full neutralization, a major part of Zn should be in the form of carbonate or oxide. The presence of carbonates might be explained by relatively high basicity ZnO, which easily adsorbs CO$_2$[32].

The UV-visible diffuse reflectance spectra of the Zn-HPW/TiO$_2$ nanocomposite and reference compounds are displayed in Fig. 3a. The sample exhibits intense absorption in the ultraviolet region (<400 nm). The band gap energy for different nanocomposites estimated using Tauc's plots[20–22] (Fig. 3b) varies from 3.0 to 3.2 eV. Relatively small effect of the promotion with Zn is observed on the band gap. The ZnO band gap energy is a function of crystallite size and varies from 3.12 to 3.30 eV[33]. Zinc carbonate also has semiconductor properties; its band gap is situated at 3.36 eV[34]. This also is rather close to the band gap of HPW ($E_g = 3.12$ eV) and TiO$_2$ ($E_g = 3.20$ eV[35]).

The TEM images of the calcined Zn-HPW/TiO$_2$ catalyst are presented in Fig. 3c. They clearly show the presence of core-shell particles. In these particles, the core is constituted by TiO$_2$ crystallites of 30–40 nm, while the shell is built by the HPW heteropolyacid. The thickness of HPW layer is about 1–2 nm. Small clusters of zinc, which is highly dispersed on the catalyst

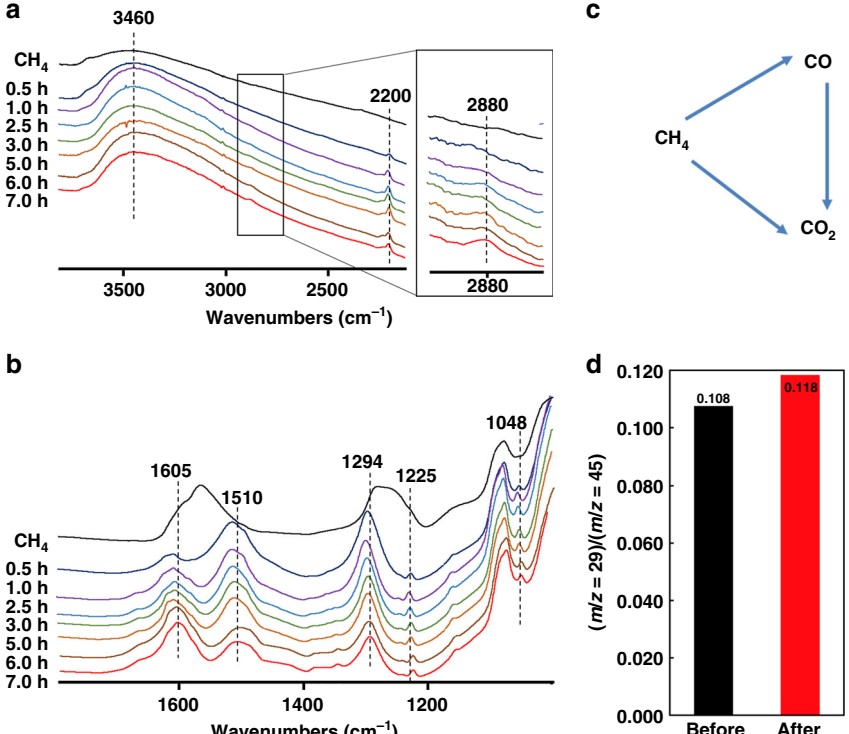

**Fig. 4** Intermediates of methane photocatalytic oxidation. **a** In situ FTIR spectra of the Zn-HPW/TiO$_2$ catalysts in the region of 3800–2100 cm$^{-1}$ and **b** 1800–1000 cm$^{-1}$. The spectra were measured under light at different reaction times in CH$_4$, **c** parallel and consequent routes in methane oxidation, **d** the ($m/z = 29)/(m/z = 45)$ ratio in the isotopic $^{13}$CO$_2$ labeling experiments: black = before and red = after photocatalytic reaction

surface, can be also observed. Figure 3d shows TEM images of used Zn-HPW/TiO$_2$. Interestingly, the methane photocatalytic oxidation does not result in any noticeable zinc sintering. Some restructuring of the HPW shell was only observed.

**Reaction paths in methane oxidation to CO over the Zn-HPW/TiO$_2$ composites.** Additional experiments were conducted to investigate in detail the reaction paths over the Zn-HPW/TiO$_2$ composites. Supplementary Fig. 5a shows variations of the CO and CO$_2$ concentrations as functions of the reaction time for methane oxidation over the Zn-HPW/TiO$_2$ catalyst, whereas the calculated CO and CO$_2$ selectivities as functions of methane conversion are displayed in Supplementary Fig. 5b. The selectivity to carbon monoxide decreases and selectivity to CO$_2$ increases as the contact time and conversion increase. The extrapolation to zero conversion gives therefore, the selectivity of primary reactions. The primary selectivity of methane oxidation to CO is about 80%, while only about 20% of methane directly oxidizes to CO$_2$. Note that some small amount of CO$_2$ can come from decomposition of surface zinc carbonate. It can be suggested that the CH$_4$-O$_2$ mixtures may react via a combination of parallel and sequential steps. Similar reaction paths were also observed for many partial oxidation reactions[36–39].

In the methane partial oxidation, CO can be formed directly from methane, while CO$_2$ is produced either from methane total oxidation or from CO oxidation (Fig. 4c).

Therefore, in order to increase the selectivity to CO, the catalyst should contain active sites, which are selective for methane direct oxidation to carbon monoxide. These sites should have high activity towards methane oxidation to CO and lower activity for CO oxidation to CO$_2$. Many reactions of partial oxidation occur with participation of oxygen atoms of the catalysts according to the Mars–van Krevelen mechanism[25]. In this mechanism, the oxygen of the catalyst first oxidizes the

molecules of substrate. Oxygen vacancies are produced on the catalyst surface. Then, the oxygen vacancies react with gaseous oxygen and the catalytic structure regenerates. To confirm the relevance of the Mars–van Krevelen mechanism for methane photo-oxidation, the following experiments were conducted.

First, the Zn-HPW/TiO$_2$ catalyst was exposed directly to methane without adding any oxygen. Figure 5a shows the concentrations of CO and CO$_2$ produced in this experiment as functions of time. The catalyst shows relatively low conversion of methane to CO and CO$_2$. Moreover, the conversion completely stops after 20 h of reaction. This probably corresponds to exhausting oxygen available in the catalyst. Assuming that the methane oxidation proceeds to carbon oxides and water with the following stoichiometry:

$$CH_4 + 3O_s \xrightarrow{h\nu} CO + 2H_2O \qquad (3)$$

$$CH_4 + 4O_s \xrightarrow{h\nu} CO_2 + 2H_2O \qquad (4)$$

Our calculation gives almost the same amount of oxygen present in ZnO in the calcined Zn-HPW/TiO$_2$ catalyst before conducting the reaction (6 wt.% Zn, $7.4 \times 10^{-4}$ mol/g$_{cat}$) and the amount of oxygen in the produced CO and CO$_2$ ($7.8 \times 10^{-4}$ mol/g$_{cat}$). This is indicative of the participation of oxygen linked to zinc in methane photocatalytic oxidation. Incorporation of zinc oxide oxygen atoms in the produced carbon monoxide and carbon dioxide suggests that zinc is reduced to the metallic state during the reaction. Indeed, the catalyst color changes from white in the calcined catalyst to grayish after the catalyst exposure to methane in the presence of light (Fig. 5a, inserts). To provide further insights into the oxidation state of zinc, both the calcined Zn-HPW/TiO$_2$ catalyst and its counterpart after exposure to

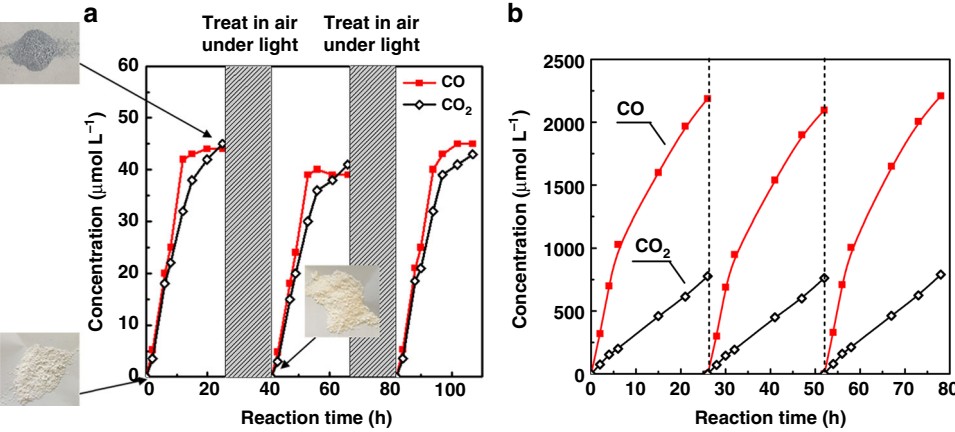

**Fig. 5** Carbon monoxide and carbon dioxide concentrations in the reactor. **a** After exposure of Zn-HPW/TiO$_2$ to pure methane (CH$_4$ 0.3 MPa). The catalyst was regenerated by treatment in 1 bar of air under light at ambient temperature. **b** After exposure of Zn-HPW/TiO$_2$ to CH$_4$ and air. Reaction conditions: catalyst, 0.1 g; gas phase pressure, CH$_4$ 0.3 MPa, Air 0.1 MPa

methane were characterized by XPS (Supplementary Fig. 6a). The exposure to methane results in a shift of the Zn 2p$_{3/2}$ lines from 1021.9 eV characteristic for Zn$^{2+}$ species to 1021.5 eV which corresponds to Zn$^0$ [40,41], This suggests zinc reduction to the metallic state. The change in the valence state from Zn$^{2+}$ to Zn$^0$ was observed even more obviously (Supplementary Fig. 6b) using the Auger spectroscopy from 3 eV downward shift of the binding energy of the Zn L$_3$M$_{4.5}$M$_{4.5}$ Auger peak[42–44], In the Zn Auger spectrum of the Zn-HPW/TiO$_2$ catalyst that was recorded after pretreatment in 0.3 Mpa of CH$_4$ under 400 W Xe lamp for 12 h, a shoulder feature appeared at a binding energy that was reduced by 3 eV. When the catalyst was regenerated in 0.1 MPa of air under 400 W Xe lamp for 12 h, the shoulder almost disappeared. Simultaneously, XPS suggests complete oxidation of Zn metal clusters after their treatment in air and under irradiation at the same conditions (Supplementary Fig. 6a). The Zn 2p$_{3/2}$ binding energy shifts from 1021.5 to 1021.9 eV, which is characteristic for zinc oxide species. Thus, exposure of the reduced Zn-HPW/TiO$_2$ catalyst in the presence of light results in zinc re-oxidation to the Zn$^{2+}$ oxidation state.

The catalytic performance can be entirely regenerated by treatment in air at room temperature in the presence of light. Figure 5a shows identical methane photocatalytic conversion on the Zn-HPW/TiO$_2$ catalyst after exposure to 0.1 MPa of air and irradiation for 16 h. The reaction-regeneration cycle can be repeated several times. Note that after exposure to air under irradiation for 16 h, color of the used catalyst again reverted from gray to white.

The second series of experiments involved catalyst simultaneous exposure to methane and air (Fig. 5b). Exposure of the pre-calcined catalyst at the same time to methane and air results in the production of both CO and CO$_2$. Similarly to the exposure of the catalyst to pure methane, the catalyst changes colors from white to grey in the end of the cycle. Note that the carbon monoxide and carbon dioxide concentrations are much higher in this experiments compared to the exposure of the catalyst only to methane (Fig. 5b). This is probably due to the presence of a larger amount of oxygen in the reactor with continuous zinc oxidation-reduction cycling during the reaction. Note that the oxygen for methane oxidation can originate both from the catalyst surface and reactor gaseous phase. The production of CO and CO$_2$ then slows down after 12 h of reaction. Slower reaction rate is probably due to the depletion of available oxygen. The calculation again suggests that the amount of oxygen atoms in the produced CO and CO$_2$ (2.4 mmol, including production of water) corresponds

almost exactly to the amount of oxygen atoms in the 0.25 L reactor (2.3 mmol). In the calculation, we considered both oxygen related to the ZnO species and oxygen atoms present as O$_2$ in the gaseous phase. The amount of the oxygen in the calcined catalysts corresponds to about 3% of the amount of the oxygen from the gaseous phase. After purging, the reactor was once again exposed to 0.3 MPa of CH$_4$ and 0.1 MPa of oxygen. Figure 5b shows that methane was again converted to CO and CO$_2$. Similar CO and CO$_2$ formation rates were observed in the second and third runs. Furthermore, the catalyst stability was maintained over the reaction period of 78 h, when the fresh reactants were re-introduced.

**In-situ FTIR study of methane photocatalytic oxidation.** The structure of the Zn-HPW/TiO$_2$ catalyst showing higher yield of CO in methane photocatalytic oxidation was studied in detail by in-situ FTIR spectroscopy. The calcined Zn-HPW/TiO$_2$ catalyst pretreated in vacuum at 250 °C exhibits FTIR bands at 1560 and 1285 cm$^{-1}$, which correspond to bidentate surface Zn carbonate (Fig. 4b). Then, the bidentate carbonate transforms into mono-dentate carbonate after 0.5 h of the reaction (FTIR bands at 1510 and 1294 cm$^{-1}$ corresponding to $\nu_{as}$(CO$_3$) and $\nu_s$(CO$_3$), respectively)[30,45]. The transformation might be explained by the reaction of bidentate carbonates with water[46]. The presence of water might be observed by high intensity of the broad FTIR band at 3460 and 1605 cm$^{-1}$ related to $\nu$(O–H) and $\delta$(H$_2$O) vibrations. Subsequent exposure to light leads to the decrease in the intensity of carbonate bands with appearance of bands at 1225 and 1048 cm$^{-1}$, which might be assigned to $\nu$(C–O) stretching bands in carbonate ester (Fig. 6) and methoxy group (CH$_3$–O) respectively[47]. The presence of methoxy groups on the catalyst surface is also confirmed by a new FTIR band of C–H stretching at 2880 cm$^{-1}$ (Fig. 4a)[47]. The bands observed at 2200 and 2300 cm$^{-1}$ (Fig. 4a) probably correspond to carbon monoxide adsorption on the low coordinated Zn$^{2+}$ ions[48,49]. It can be suggested that these unsaturated Zn$^{2+}$ ions may play an important role in methane activation. Strong methane chemisorption over smaller clusters of zinc oxide has been observed by several groups[50–53]. During the catalyst regeneration, a gradual decrease in the intensity of the bands assigned to adsorbed CO species with simultaneous increase in the intensity of the bands assigned to carbonates was observed.

FTIR analysis of the gaseous phase (Supplementary Fig. 7) clearly shows the presence of methane in the IR cell at the initial periods of the reaction (CH rotation-stretching and rotation-bending bands at

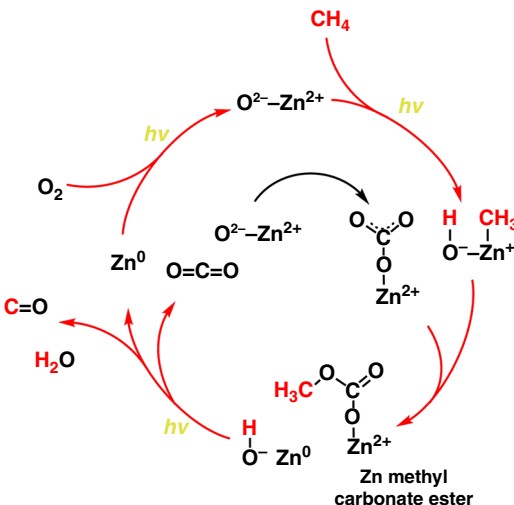

**Fig. 6** Reaction steps in methane photo-oxidation over Zn-HPW/TiO$_2$

around 3020 and 1300 cm$^{-1}$, respectively). At longer reaction time, gaseous carbon monoxide was identified by rotation- stretching bands at 2150 cm$^{-1}$. In agreement with the results of photocatalytic tests, the intensity of the carbon monoxide FTIR band at 2150 cm$^{-1}$ increases with the reaction time.

In order to confirm the presence of zinc methyl carbonate ester, dimethyl carbonate [(CH$_3$O)$_2$CO, DMC] was adsorbed on the Zn-HPW/TiO$_2$ catalyst with simultaneous measurements of IR spectra. The set of bands similar to those observed during reaction has been observed (Supplementary Fig. 8a). Treatment of the catalyst with adsorbed DMC in light leads to a gradual decrease in the intensity of the DMC bands and increase in the water bending band at 1605 cm$^{-1}$. FTIR analysis of gas phase shows also an increase in the intensity of CO$_2$ and CO bands, which indicates decomposition of DMC in the presence of light.

The Zn-HPW/TiO$_2$ catalyst was also exposed to the DMC vapor in the photocatalytic reactor in parallel experiments. In the absence of light, small amount of CO$_2$ was produced, which is probably due to the DMC hydrolysis. Interestingly, in the presence of light, photocatalytic decomposition of DMC leads to the production of mostly CO (Supplementary Fig. 9).

Isotopic labeling experiments were performed in order to provide further information about the reaction mechanism. The experiments were conducted under a $^{12}$CH$_4$, O$_2$ and $^{13}$CO$_2$ atmosphere (0.3 MPa of CH$_4$, 0.1 MPa of O$_2$ and 1% isotopic $^{13}$CO$_2$). The goal was to elucidate if CO$_2$ from the gaseous phase can be involved in the reaction. In these experiments we clearly observed an increase in the $^{12}$CO ($m/z = 28$) and $^{12}$CO$_2$ ($m/z = 44$) signals after the reaction relative to the $^{12}$CH$_4$ ($m/z = 16$) signal (Supplementary Fig. 10). This suggests that $^{12}$CH$_4$ was converted to $^{12}$CO and $^{12}$CO$_2$. Figure 4d displays the ($m/z = 29$)/($m/z = 45$) ratio before (black) and after (red) photocatalytic reaction and clearly indicates a significant (+10%) increase. This increase could owe to the conversion of $^{13}$CO$_2$ to $^{13}$CO under the reaction conditions.

## Discussion

It is well known that the absorption of a photon corresponding to the fundamental absorption band of an oxide leads to the formation of electron and hole pairs, i.e., excitons, which undergo radiative decay[54]. For the Zn–O sites, this process is represented as [Zn$^{2+}$–O$^{2-}$] ↔ [Zn$^+$–O$^-$].

This process corresponds to the band gap transition in zinc oxide with the energy of 3.2 eV. The photocatalytic activity of the

supported metal oxides is therefore closely associated with the charge-transfer excited complex [Zn$^+$–O$^-$] formed on the surface. This suggestion is also consistent with the uncovered dependence of the methane photo-oxidation rate on the irradiation wavelength. The reaction rate increases almost twenty times, when the catalyst has been exposed to UV irradiation compared to the exposure to visible light. In the presence of the UV irradiation, the [Zn$^+$–O$^-$] complex would favor activation and homolytic dissociation of methane molecules. Our experiments show that methane photo-oxidation occurs with participation of oxygen from the catalyst. Indeed, direct exposure of the oxidized catalysts to methane results in production of mainly CO, while the zinc is being reduced to metallic state. These reduced zinc particles can then easily reoxidized and the catalyst regenerated.

Previously, participation of the oxygen from the catalyst lattice in the photo-oxidation of methylene blue was shown by Ali et al.[26,27] over deposited ZnO thin films. The Mars–van Krevelen type mechanism was also observed by Lee and Falconer[55] in photocatalytic decomposition of formic acid on TiO$_2$. Oxygen atoms for oxidation of formic acid were extracted from the TiO$_2$ lattice. The Mars–van Krevelen oxidation–reduction mechanism commonly operates for many reactions of selective and partial oxidation. In agreement with these previous reports, our results also indicate an important role of the Mars–van Krevelen mechanism in the reactions of methane partial photocatalytic oxidation, which target selective production of carbon monoxide instead of CO$_2$.

The concentration of zinc carbonate also significantly decreases during the methane oxidation. The in situ FTIR data are indicative of the formation of the surface methyl carbonate during the reaction. The conducted experiments with DMC adsorbed on the catalysts suggest that CO is a major product of decomposition of methyl carbonate in the presence of light. The following reaction sequence (Fig. 6) is proposed to interpret the obtained photocatalytic and spectroscopic data. The first stage of the reaction is methane activation via its dissociation over Zn–O pairs under irradiation followed by the formation of Zn-methyl species. The reaction of surface Zn-methyl species with carbonate results in formation of the surface methyl carbonates. The surface methyl carbonates were identified by the C–O stretching bands at 1225 cm$^{-1}$ in the carbonate ester and C–O and C–H stretching bands at 1048 and 2880 cm$^{-1}$, respectively, in methoxy fragments. At the same time, zinc is reduced to the metallic state. In agreement with the DMC adsorption experiments, decomposition of zinc methyl carbonate leads to carbon monoxide. The participation of zinc carbonates in the reaction is consistent with isotope tracing experiments. Indeed, addition of small amounts of $^{13}$CO$_2$ to the reacting medium results in the appearance of $^{13}$CO. This suggests that during methane partial photo-oxidation, CO is formed via decomposition of the surface species such as methyl carbonates produced from zinc carbonates.

Thus, the obtained results advocate in favor of the extremely important role of Zn–O pairs in the composite Zn-HPW/TiO$_2$ catalysts in methane photocatalytic oxidation to CO. These pairs are involved in the methane dissociation and formation of carbonate and methyl carbonate species. They can be associated either with highly dispersed ZnO clusters or with at least two Zn$^{2+}$ ions in the cationic sites of HPW.

Carbon dioxide also observed in methane photo-oxidation can be produced either directly from methane or though methane consecutive oxidation to CO and then to CO$_2$ (Fig. 4c) On TiO$_2$ and HPW, a significant fraction of CO$_2$ is probably produced via direct methane oxidation, while over Zn-HPW/TiO$_2$, CO$_2$ seems to primarily occur via CO secondary oxidation. CO$_2$ is mostly produced as a secondary product of methane photo-oxidation over the zinc-containing catalysts. The mechanism of the CO

total oxidation to $CO_2$ over $TiO_2$ has been investigated in several previous reports[56–58]. Linsebigler et al.[57,58] have reported that the reaction proceeds over on vacancy defect sites on the $TiO_2$ surface and involves $O_2^-$ surface species.

To summarize, among metal-tungstophosphoric composite catalysts supported on titania, the zinc counterpart exhibits remarkable activity in direct methane photocatalytic oxidation at ambient temperature with extremely high selectivity to carbon monoxide. In the composite catalysts, tungstophosphoric acid constitutes a thin layer of 1–2 nm over titanium oxide, while zinc species which are involved in methane photocatalytic oxidation, are highly dispersed on the tungstophosphoric layer.

The methane photocatalytic oxidation proceeds as a combination of parallel and consecutive reactions with carbon monoxide being a primary reaction product. The carbon monoxide yield from methane can reach 3–4% and give high QE (7.1% at 362 nm) values under the optimized reaction conditions. The reaction is consistent with the Mars–van Krevelen type oxidation–reduction sequence and involves formation of the surface methyl carbonate as reaction intermediate and zinc oxidation–reduction cycling.

## Methods

**Chemicals.** Titanium (IV) oxide (P25, $TiO_2$, 99.5%), phosphotungstic acid hydrate ($H_3O_{40}PW_{12}\cdot xH_2O$, $M_w = 2880.05$), zinc nitrate hexahydrate ($Zn(NO_3)_2\cdot6H_2O$, ≥99.0%), iron(III) nitrate nonahydrate ($Fe(NO_3)_3\cdot9H_2O$, ≥98%), ammonium metavanadate(V) ($NH_4VO_3$, 99%), cobalt(II) nitrate hexahydrate ($Co(NO_3)_2\cdot6H_2O$, 98%), cerium(III) nitrate hexahydrate ($Ce(NO_3)_3\cdot6H_2O$, 99%), gallium(III) nitrate hydrate ($Ga(NO_3)_3\cdot xH_2O$, 99.9%), copper(II) nitrate trihydrate ($Cu(NO_3)_2\cdot3H_2O$, 99–104%), silver nitrate ($AgNO_3$, ≥99.0%), palladium(II) nitrate hydrate ($Pd(NO_3)_2\cdot xH_2O$, 99.9%) and dimethyl carbonate (DMC, ≥99.0%) were purchased from Sigma-Aldrich and used without further purification. $^{13}C$ labeled carbon dioxide ($^{13}CO_2$, 99 atom% $^{13}C$ - <3 atom% $^{18}O$) was purchased from Cortecnet.

**Synthesis of the metal-HPW/TiO2 composite catalysts.** $TiO_2$ (P25) constituted by 20% rutile and 80% anatase was used as catalytic support. The metal-HPW/$TiO_2$ catalysts were prepared by the two-step impregnation. For example, for preparation of Zn-HPW/$TiO_2$, first, a fixed amount of $TiO_2$ was suspended in an anhydrous ethanol solution of phosphotungstic acid hydrate ($H_3[P(W_3O_{10})_4]$ x$H_2O$, HPW). The ratio of HPW to $TiO_2$ varied from 0.15 to 0.6. The HPW/$TiO_2$ (HPW/$TiO_2$ ratio = 0.3) sample was obtained by stirring and drying at 353 K for 12 h. Second, the Zn-HPW/$TiO_2$ catalyst was prepared by incipient wetness impregnation of the support with aqueous solutions of zinc (II) nitrate hexahydrate ($Zn(NO_3)_2\cdot6H_2O$). For the other metal-HPW/$TiO_2$ catalysts, the aqueous solutions of $Fe(NO_3)_3\cdot9H_2O$, $NH_4VO_3$, $Co(NO_3)_2\cdot6H_2O$, $Ce(NO_3)_3\cdot6H_2O$, $Ga(NO_3)_3\cdot xH_2O$, $Cu(NO_3)_2\cdot3H_2O$, $AgNO_3$ and $Pd(NO_3)_2\cdot xH_2O$ were used. The concentrations of the impregnating solutions were calculated in order to obtain about 6 wt.% metal in the final catalysts. After the second impregnation, the catalysts were dried overnight in an oven at 373 K. Then, they were calcined in air at 300 °C for 3 h with 2 °C min$^{-1}$ temperature ramping. The catalysts were denoted as M-HPW/$TiO_2$, and M = V, Fe, Ga, Ce, Co, Cu, Ag, Pd, and Zn.

**Characterization.** The X-ray powder diffraction (XRD) experiments were conducted using a Bruker AXS D8 diffractometer with Cu K$_\alpha$ radiation ($\lambda = 0.1538$ nm). The XRD patterns were collected in the 5–80° (2 $\theta$) range. The diffuse reflectance UV-visible spectra of the catalysts were recorded on a Perkin-Elmer Lambda 650S UV/VIS spectrometer that was equipped with an integrating sphere covered with $BaSO_4$ as reference. The TEM observations of the samples were performed on a Tecnai instrument, equipped with a LaB6 crystal, operating at 200 kV. The point resolution was around 0.24 nm. Prior to the analysis, the samples were dispersed by ultrasound in ethanol solution for 5 min, and a drop of solution was deposited onto a carbon membrane onto a 300 mesh-copper grid. FTIR spectra have been collected using a Thermo iS10 spectrometer at a 4 cm$^{-1}$ resolution (0.96 cm$^{-1}$ data spacing). The spectra were analyzed and presented (including integration, differentiation and determination of peak positions) using a specialized Thermo software (Omnic). The XPS spectra were taken using a Kratos Axis spectrometer, equipped with an aluminum monochromater for a 1486.6 eV source working at 120 W. The powder samples were pressed into 6 mm diameter pellets. The binding energies were corrected with respect to C 1s of 284.6 eV and the binding energies were estimated within ±0.2 eV.

**Photocatalytic tests.** The photocatalytic conversion of methane was carried out in a homemade stainless-steel batch reactor (volume, ~250 mL) with a quartz window

on the top of the reactor (Supplementary Fig. 11). The light source was 400 W Xe lamp (Newport). The Quantum Efficiency (QE) was determined by using a Hamamatsu spot light sources LC8-06 Hg-Xe lamps emitting between 240 and 600 nm and equipped with a quartz light-guide to deliver a stable and uniform illumination of the sample. The spectral range of the irradiation where selected by using Hamamatsu optical filter. The irradiance measured by an optical power meter (Newport PMKIT) were 94 and 38 mW cm$^{-2}$ in the ranges of >382 and 280–400 nm, respectively. All the photocatalytic tests were performed at ambient temperature. The following typical procedure was used. First, 0.1 g of solid catalyst was placed on a quartz glass holder on the bottom of reactor. Then, the reactor was filled with $CH_4$. The methane pressure was increased up to 0.3 MPa. The reactor contained 0.3 MPa of $CH_4$ and 0.1 MPa of air (78% $N_2$ and 21% $O_2$). Before starting a photocatalytic reaction, the reactor was kept in the dark for 1 h to ensure an adsorption-desorption equilibrium between the photocatalyst and reactants. Subsequently, the reactor was irradiated by a 400 W Xe lamp. The photocatalytic reaction time was typically 6 h and varied from 1 to 50 h. The reaction products (CO and $CO_2$) were analyzed by gas chromatography (GC, PerkinElmer Clarus® 580). The reaction system was connected to an online GC injection valve, and the gaseous products were directly introduced to the GC for analysis. The GC was equipped with a PoraBOND Q, a ShinCarbon ST 100/120, columns, a flame ionization detector (FID) and a thermal conductivity detector (TCD). Helium was used as carrier gas.

**$^{13}CO_2$ labeling experiment.** The isotopic $^{13}CO_2$ labeling experiments were performed in the homemade stainless-steel batch reactor. First, 0.1 g of solid catalyst was placed on a quartz glass holder on the bottom of reactor. Then, the reactor was sequentially filled with $O_2$, which was regulated to 0.1 MPa, $CH_4$ and $^{13}CO_2$. The reactor contained 0.3 MPa of $CH_4$, 0.1 MPa of $O_2$ and 0.004 MPa isotopic $^{13}CO_2$. The reactor was kept at ambient temperature. Before starting a photocatalytic reaction, the reactor was kept in the dark for 1 h to ensure an adsorption–desorption equilibrium between the photocatalyst and reactants and the gas phase was analyzed by mass spectrometry. In particular, the ($m/z = 29$)/($m/z = 45$) was recorded to determine the contribution of $^{13}CO_2$ ($m/z = 45$) cracking to $^{13}CO^+$ ($m/z = 29$) within the ion source of the mass spectrometer. Subsequently, the reactor was irradiated by a 400 W Xe lamp for 14 h. After reaction, the isotopic products ($^{13}CO$ and $^{13}CO_2$) were again analyzed by recording ($m/z = 29$)/($m/z = 45$) ratio and compared to the value obtained before reaction. Any increase of the ($m/z = 29$)/($m/z = 45$) ratio would be indicative of the formation of $^{13}CO$ in the reactor vessel through the conversion of $^{13}CO_2$. To perform gas phase analysis, the gases from the reaction vessel were slowly released in a 10 ml min$^{-1}$ He flux and analyzed using an online mass spectrometer (Omnistar GSD300 from Pfeiffer Vacuum).

**Measurement of quantum efficiency.** We measured quantum efficiency (QE) at 362 nm for the photocatalytic conversion of $CH_4$ over the 6 wt.% Zn-HPW/$TiO_2$. The quantum efficiency ($\eta$) for the formation of a product was calculated using the following equation:

$$\eta = \frac{R(electron) \times N_A}{I(W\,cm^{-2}) \times S(cm^2) * t(s)/E_\lambda(J)} \times 100\% \qquad (5)$$

Where $N_A$, $I$, $S$, $t$ represents the Avogadro's constant, light irradiance on the sample, irradiation area and reaction time, respectively. $E_\lambda$ is given by $hc/\lambda$ ($\lambda = 362$ nm). $R(electron)$ represents the amounts of electrons used in the formation of the product. $R_{CO}(electron)$ are the amounts of electrons used for the formations of CO. $R_{CO}(electron) = 6n(CO)$, where $n$ is the amount of CO product.

## Data availability

The data that support the findings of this study are available from the corresponding author upon reasonable request.

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

## Acknowledgements

We thank Laurence Burylo, Pardis Simon, and Martine Frère for help with XRD and XPS measurements. X.Y. thanks the Chinese scholarship council for providing him a stipend for PhD studies in France. Chevreul Institute (FR 2638), Ministère de l'Enseignement Supérieur, de la Recherche et de l'Innovation, Hauts-de-France Region and FEDER are acknowledged for supporting and funding partially this work.

## Author contributions

X.Y., V.V.O. and A.Y.K. conceived the idea for this work. All authors contributed to the design of the experimental setup and experimental procedures. X.Y. prepared the catalysts, collected the data, and performed ex-situ characterization. X.Y. and V.D.W. per-

formed evaluation of quantum efficiency. A.L. conducted isotopic labeling experiments. V.V.O. and X.Y. performed in-situ FTIR experiments. X.Y. and A.Y.K. wrote the draft and all authors took part in improving the manuscript.

## Additional information

**Competing interests:** The authors declare no competing interests.

