## [Peer Review File · Nature Communications]

Reviewers' comments:

Reviewer #1 (Remarks to the Author):

The manuscript describes the high selectivity of methane oxidation to carbon monoxide over zinc heteropolyacid-titania photocatalyst. The mechanism has been investigated and proposed in this work, including the key intermediate for the reaction. The authors claimed that the work is the first example of utilizing photocatalysis for selective conversion of methane into carbon monoxide, which is not true since another type of photocatalysis is also possible to obtain carbon monoxide as the sole oxidized product via photocatalytic dry reforming of methane. The authors also stated that the current yield (3-4%) of carbon monoxide is practical for industrial application. This is not convincing since in my opinion, it is still far from the real one. While the findings might be interesting for those working in photocatalysis, it might not for other disciplines. Furthermore, there are some insufficient discussions which need to be clarified by the authors, especially for the active and selective sites. Therefore, in my opinion, the work shall be completed first to give a clearer conclusion and the work shall be submitted to a more specific journal focusing on the (photo)catalysis.

Some comments are also given below for further improvement:

1. In the introductory part, the works about photocatalytic dry reforming of methane shall be also introduced since the oxidized product could be also carbon monoxide alone, and thus the methane is selectively converted to carbon monoxide.
2. As shown in Figure S1, the ratio of HPW/TiO₂ clearly affected the rate of CO, but not much for CO₂ formation rate. This part shall be discussed.
3. It was stated that Zn species was the important site for CO selectivity. However, Figure 2 showed different results. The CO₂ formation rate was not much changed by the Zn species, but the formation of CO was affected by Zn content. Why was the increase of CO only observed at 2-6 wt%? The formation of CO decreased when the Zn content was low or high. This could be related to the different Zn species formed on these samples, which shall be clarified.
4. As shown in Figure 3, the activities and selectivities of TiO₂, HPW, and HPW/TiO₂ are very similar to each other. Even though HPW might have semiconductor properties, its similar activity to active TiO₂ might raise questions and shall be discussed clearly.
5. Still in Figure 3, while the addition of Zn seemed to slightly increase the activity of HPW and TiO₂, it did not give increased selectivity for CO formation. In contrast, the Zn-HPW/TiO₂ gave much higher activity and selectivity than the Zn/HPW and ZnO/TiO₂. The authors mentioned that the Zn species are the selective sites for CO formation, but how it could not be observed on the Zn/HPW? What are the differences in the structure and properties of the Zn species in these two samples?
6. Based on the current data, the authors could not give clear suggestions about the active species on the Zn-HPW/TiO₂. The proposed species active could be ZnO, clustered Zn atoms, Zn²⁺ cations, or even small possible charged Zn cationic nanoclusters. It would be better to have a clearer conclusion after some clarifications are made (such as shown in points 3 and 5 above).

Reviewer #2 (Remarks to the Author):

The conversion of methane under mild conditions remains a challenge task. The authors reported in this paper by preparing a composite catalyst based on zinc, tungstophosphoric acid (HPW) and titania, methane can be photocatalytically oxidized into carbon monoxide at ambient conditions. In-situ FTIR and XPS spectra were further employed to investigate the photocatalytic mechanism. The catalytic performance was suggested to be related to the zinc species dispersed on HPW/titania, and, the reaction proceeds via formation of surface methoxy carbonates as key reaction intermediates. The results is interesting and the manuscript was well organized and clearly written. However, the below deficiencies make the manuscript cannot be accepted for publication at current a stage:

1. The authors used Xe lamp as the light source to do the photocatalytic experiments. The catalyst used seems only harvest UV light. It is strongly suggested the authors do more experiments to demonstrate the influence of UV, visible and IR light on the methane conversion.
2. Can the authors provide Q.E of the photocatalytic methane conversion?
3. The mechanism discussion involves too much speculation. If isotopic labeling can be conducted the conclusions would be more reliable.
4. The durability test under flow gas mode is strongly encouraged.
5. The mechanism discussion neglected the by-product.

Reviewer #3 (Remarks to the Author):

General points:

I have read a few times the Ms and the opinion about this study is stated below.

The experiments reported seem feasible and the results reported possible. The Ms is of general interest for the readership working in the area of catalytic conversion of available materials to valuable chemical products. The English used is appropriate as well as the techniques to evaluate the reaction and the surface of the catalyst used. The authors report the selective methane oxidation at ambient conditions to obtain CO on Zn-polytungstate-titania in a reaction that seems to follow the Mars-Krevelen mechanism. This work reports the light induced conversion of CH₄ into CO with marginal CO production, and this is an interesting observation.

Specific points:

p1: Abstract: Delete cut the first 5 lines, they do not condense the results obtained during the course of this study.

p.3-5: Introduction: Eliminate the prehistory of the problem at hand, cut the introduction by 40% cite items directly related to the problem at hand, beginning after reference 24.

p.8: Figure 4 move to support material, delete the text write-up. This Figure is not important in the context of the study.

p.10: Fig. 7a/7b move to support material, delete the text write-up. This Figure is also not important for the continuity of exposition of the Ms-material.

p.10-14 Discussions. Make them more pointed and precise cut text by 1/3.

p.17-17: Delete some details in the FTIR section, to many details are given. Concentrate in the main results.

p.17/19: Discussion: It is dispersed. Group 4 o 5 points of interests and condense the material cutting 1/4 of the actual section in the Ms.

p.20: Delete last paragraph of the Ms lines 400-411. Speculative and the authors know this.

Responses to the Referees' comments

Reviewer #1 (Remarks to the Author):

Comment: The manuscript describes the high selectivity of methane oxidation to carbon monoxide over zinc heteropolyacid-titania photocatalyst. The mechanism has been investigated and proposed in this work, including the key intermediate for the reaction. The authors claimed that the work is the first example of utilizing photocatalysis for selective conversion of methane into carbon monoxide, which is not true since another type of photocatalysis is also possible to obtain carbon monoxide as the sole oxidized product via photocatalytic dry reforming of methane. The authors also stated that the current yield (3-4%) of carbon monoxide is practical for industrial application. This is not convincing since in my opinion, it is still far from the real one. While the findings might be interesting for those working in photocatalysis, it might not for other disciplines. Furthermore, there are some insufficient discussions, which need to be clarified by the authors, especially for the active and selective sites. Therefore, in my opinion, the work shall be completed first to give a clearer conclusion and the work shall be submitted to a more specific journal focusing on the (photo)catalysis.

Reply and revision

We thank Reviewer 1 for careful reading of our manuscript and constructive comments. Please find below replies to the comments and modifications made in the text of manuscript.

Some comments are also given below for further improvement:

Comment 1. In the introductory part, the works about photocatalytic dry reforming of methane shall be also introduced since the oxidized product could be also carbon monoxide alone, and thus the methane is selectively converted to carbon monoxide.

Reply

In the revised manuscript, we added several references to recent papers relevant to the photocatalytic methane dry reforming. Indeed, methane dry reforming represents an alternative and interesting route for production of carbon monoxide. Methane dry photothermal reforming usually involves both thermo- and photo-catalysis. It conducted at relatively high temperatures (at least 200°C or even higher, see Refs 17 and 18). Note that in our work, methane selective photooxidation to carbon monoxide occurs at ambient temperature.

Revision

P. 3. “A limited number of papers [17, 18] also addressed combined photo-thermocatalytic [19] or plasma-enhanced [20] methane dry reforming, which represents an interesting route for production of carbon monoxide.”

P. 5-6 .“The results of methane partial oxidation to CO can be compared with the results of methane dry reforming. Note that in our work, methane photooxidation to carbon monoxide occurs with high selectivity at ambient temperature, while in previous reports [17, 18], methane dry reforming was conducted at relatively high temperatures in order to obtain noticeable conversion. Methane dry reforming usually involves both thermo- and photo-catalysis.”

Comment 2. As shown in Figure S1, the ratio of HPW/TiO₂ clearly affected the rate of CO, but not much for CO₂ formation rate. This part shall be discussed.

Reply. The effect of HPW/TiO₂ ratio on the rate of CO production was not very significant. This issue has been addressed more clearly in the revised manuscript.

Revision: P. 7 *“The HPW/TiO₂ ratio in the composite Zn-HPW/TiO₂ catalysts does not noticeably affect the rate of CO₂ formation, while the effect of this ratio on the rate of CO formation is more significant. Note that only the CO production rate is strongly influenced by the concentration of highly dispersed Zn species. It is expected that higher HPW/TiO₂ ratio could enhance zinc dispersion, because of possible localization of zinc ions in the cationic sites of HPW. “*

Comment 3. It was stated that Zn species was the important site for CO selectivity. However, Figure 2 showed different results. The CO₂ formation rate was not much changed by the Zn species, but the formation of CO was affected by Zn content. Why was the increase of CO only observed at 2-6 wt%? The formation of CO decreased when the Zn content was low or high. This could be related to the different Zn species formed on these samples, which shall be clarified.

Reply: *Yes, indeed. The manuscript was modified to address this issue. Figure 2 shows that the CO formation rate significantly increases (from 100 to more than 400 $\mu\text{mol g}_{\text{cat}}^{-1} \text{h}^{-1}$) as a function of Zn content, while the rate of CO₂ production is affected to a much lesser extent by the catalyst promotion with zinc. Note that both CO and CO₂ are also produced with very low rates on TiO₂, HPW, HPW/TiO₂ etc. Addition of Zn mostly increases the rate of CO production. The*

rate of CO production mostly depends on the concentration of zinc species. Because of lower Zn dispersion, the CO production rate decreases at higher zinc content in the catalysts.

Revision P. 6 *“Thus, formation of CO₂ might be explained by the activity of HPW/TiO₂, while the Zn species seem to be active and selective in methane photo-oxidation to CO. Another important observation is that the major increase in CO is only observed when the Zn content is higher than 2-3 wt. %. The highest rate of methane oxidation was observed at Zn content of 6.0 wt. %. Note that the activity somehow decreases at higher Zn loadings probably because of lower zinc dispersion. The highest zinc dispersion can be obviously obtained at lower zinc content, at the amount of Zn²⁺ ions, which can neutralize the acid hydroxyl groups in the HPW heteropolyacid. The maximum amount of Zn²⁺ ions necessary to neutralize acid sites corresponds to about 2 wt. % Zn in the catalyst. “*

Comment 4. As shown in Figure 3, the activities and selectivities of TiO₂, HPW, and HPW/TiO₂ are very similar to each other. Even though HPW might have semiconductor properties, its similar activity to active TiO₂ might raise questions and shall be discussed clearly.

Reply. The activity of TiO₂, HPW, and HPW/TiO₂ was 10-20 times lower than that of the Zn/HPW-TiO₂ catalyst. The selectivity was also very different. The follow paragraph was added to the manuscript to explain these differences.

Revision P. 7. *“TiO₂, HPW, and HPW/TiO₂ exhibit some activity in methane photo-oxidation. The activity was at least 10-20 times lower than over the Zn/HPW-TiO₂ and the selectivity pattern was also very different. The selectivity of methane photo-oxidation on TiO₂, HPW, and HPW/TiO₂ primarily results in CO₂, while CO was the major product over Zn/HPW-TiO₂. This*

could suggest different mechanism and kinetics of methane photo-oxidation. The lattice oxygen activated by photo-generated hole could be the main active species for the activation of methane and oxygen and subsequent oxidation of the CH₃ radicals to CO₂ over those semiconductors [10].”

Comment 5. Still in Figure 3, while the addition of Zn seemed to slightly increase the activity of HPW and TiO₂, it did not give increased selectivity for CO formation. In contrast, the Zn-HPW/TiO₂ gave much higher activity and selectivity than the Zn/HPW and ZnO/TiO₂. The authors mentioned that the Zn species are the selective sites for CO formation, but how it could not be observed on the Zn/HPW? What are the differences in the structure and properties of the Zn species in these two samples?

Reply. Indeed, we attributed high selectivity in methane photooxidation to the presence of highly dispersed Zn²⁺ species. These species only exist in the composite catalysts containing simultaneously TiO₂, HPW and zinc. Zn/TiO₂ and Zn/HPW show only relatively low activity in the CH₄ photooxidation with CO₂ being the major product. The reasons of the poor catalytic performance of Zn/TiO₂ and Zn/HPW in methane photo-oxidation to CO are given below.

Revision: P. 8. *“Promotion of pure TiO₂ or HPW with Zn results only in a slight increase in the methane oxidation rate compared to the pristine semiconductors, whereas CO₂ remains the major reaction product. The mediocre catalytic performance of those composites can be due to the following phenomena. First, Zn/TiO₂ contains relatively large ZnO crystallites detected by XRD. In this catalyst therefore, zinc has relative lower dispersion. In addition, because of poor zinc dispersion, a significant part of the TiO₂ surface is uncovered by zinc. This leads to an important contribution of the TiO₂ surface sites to methane total oxidation to CO₂. The HPW*

heteropolyacid plays a crucial role in enhancement of zinc dispersion. Indeed, TEM images (Figure 6) suggest the presence of extremely small Zn clusters in the composite Zn-HPW/TiO₂ catalyst. In addition, HPW could be efficient in transfer of holes and electrons from TiO₂ to Zn sites [29]. Second, TiO₂ is a semiconductor, which is very efficient in light harvesting and charge separation. In the absence of TiO₂, the reaction rate is low on Zn-HPW, principally because of low light harvesting. Even redispersion of Zn-HPW over silica does not lead to higher rate of methane photo-oxidation to CO (Figure S2, SM). “

Comment 6. Based on the current data, the authors could not give clear suggestions about the active species on the Zn-HPW/TiO₂. The proposed species active could be ZnO, clustered Zn atoms, Zn²⁺ cations, or even small possible charged Zn cationic nanoclusters. It would be better to have a clearer conclusion after some clarifications are made (such as shown in points 3 and 5 above).

Revision: P.20 *“The obtained results advocate in favor of the extremely important role of Zn-O pairs in the composite Zn-HPW/TiO₂ catalysts in methane photocatalytic oxidation to CO. These pairs are involved in the methane dissociation and formation of carbonate species. These Zn-O pairs can be associated either with highly dispersed ZnO clusters or with at least two Zn²⁺ ions in the cationic sites of HPW. “*

The mechanistic details are given on P. 19-20.

Reviewer #2 (Remarks to the Author):

The conversion of methane under mild conditions remains a challenge task. The authors reported in this paper by preparing a composite catalyst based on zinc, tungstophosphoric acid (HPW) and titania, methane can be photocatalytically oxidized into carbon monoxide at ambient conditions. In-situ FTIR and XPS spectra were further employed to investigate the photocatalytic mechanism. The catalytic performance was suggested to be related to the zinc species dispersed on HPW/titania, and, the reaction proceeds via formation of surface methoxy carbonates as key reaction intermediates. The results is interesting and the manuscript was well organized and clearly written. However, the below deficiencies make the manuscript cannot be accepted for publication at current a stage:

Reply and revision.

We thank Reviewer 2 for the thorough evaluation of our manuscript and useful comments. Below please find our replies, modifications and description of conducted experiments.

Comment 1. The authors used Xe lamp as the light source to do the photocatalytic experiments. The catalyst used seems only harvest UV light. It is strongly suggested the authors do more experiments to demonstrate the influence of UV, visible and IR light on the methane conversion.

Revision: P. 5-6 *“In order to evaluate the influence of UV, visible and IR light on the methane conversion, we conducted photocatalytic experiments on selected spectral ranges ($280 < \lambda < 400$ nm and $\lambda > 380$, **Table S1, SM**). The results show that the catalyst is very sensitive to the irradiance spectral range. The catalyst provides only very mild activity under visible irradiation,*

which it does not absorb, while the reaction rate increases 20 times, when the reactor is exposed to UV”

P. 18-19 “This process corresponds to the band gap transition in zinc oxide with the energy of 3.2 eV. The photocatalytic activity of the supported metal oxides is therefore closely associated with the charge-transfer excited complex $[Zn^+-O^-]$ formed on the surface. This suggestion is also consistent with the uncovered dependence of the methane photo-oxidation rate on the irradiation wavelength. The reaction rate increases almost twenty time, when the catalyst has been exposed to UV irradiation compared to the exposure to visible light. “

Comment 2. Can the authors provide Q.E of the photocatalytic methane conversion?

Revision: P. 26 “Measurement of quantum efficiency

We measured quantum efficiency(QE) at 362 nm for the photocatalytic conversion of CH₄ over the 6wt. % Zn-HPW/TiO₂. The apparent quantum yield (η) for the formation of a product was calculated using the following equation:

$$\eta = \frac{R(\text{electron}) * N_A}{I(W/cm^2) * S(cm^2) * t(s)/E_\lambda(J)} * 100\%$$

Where N_A , I , S , t represents the Avogadro’s constant, light irradiance on the sample, irradiation area and reaction time, respectively. E_λ is given by hc/λ ($\lambda = 362$ nm). $R(\text{electron})$ represents the amounts of electrons used in the formation of the product. $R_{CO}(\text{electron})$ are the amounts of electrons used for the formations of CO. $R_{CO}(\text{electron}) = 6n(\text{CO})$, where n is the amount of CO product.”

The apparent quantum yield (AQY) was 7.1% at 362 nm.

Comment 3. The mechanism discussion involves too much speculation. If isotopic labeling can be conducted the conclusions would be more reliable.

Reply. We confirmed that CO production in methane photocatalytic oxidation involved surface zinc carbonates. The results are presented in **Figure 9** and **Figure S13, SM**.

Revision: P. 18 *“Isotopic labelling experiments were performed in order to provide further information about the reaction mechanism. The experiments were conducted under a $^{12}\text{CH}_4$, O_2 and $^{13}\text{CO}_2$ atmosphere (0.3 MPa of CH_4 , 0.1 MPa of O_2 and 1% isotopic $^{13}\text{CO}_2$). The goal was to elucidate if CO_2 from the gaseous phase can be involved in the reaction. In these experiments we clearly observed an increase in the ^{12}CO ($m/z = 28$) and $^{12}\text{CO}_2$ ($m/z = 44$) signals after the reaction relative to the $^{12}\text{CH}_4$ ($m/z = 16$) signal (**Figure S13, SM**). This suggests that $^{12}\text{CH}_4$ was converted to ^{12}CO and $^{12}\text{CO}_2$. **Figure 9** displays the ($m/z=29$)/($m/z=45$) ratio before (black) and after (red) photocatalytic reaction and clearly indicates a significant (+10%) increase. This increase could owe to the conversion of $^{13}\text{CO}_2$ to ^{13}CO under the reaction conditions. “*

P. 25 *“ $^{13}\text{CO}_2$ labeling experiment*

The isotopic $^{13}\text{CO}_2$ labeling experiments were performed in the homemade stainless-steel batch reactor. First, 0.1 g of solid catalyst was placed on a quartz glass holder on the bottom of reactor. Then, the reactor was filled with O_2 , which was regulated to 0.1 MPa, and the reactor was filled with CH_4 and $^{13}\text{CO}_2$. The reactor contained 0.3 MPa of CH_4 , 0.1 MPa of O_2 and 1% isotopic $^{13}\text{CO}_2$. The temperature of the reactor was kept at ambient temperature. Before starting a photocatalytic reaction, the reactor was kept in the dark for 1 h to ensure an adsorption-desorption equilibrium between the photocatalyst and reactants. Subsequently, the reactor was irradiated by a 400 W Xe lamp for 14 h. The isotopic products (^{13}CO and $^{13}\text{CO}_2$) were analyzed

by mass spectrometer. The reaction system was connected to an online mass spectrometer analyzer (Omnistar GSD300 from Pfeiffer Vacuum). The gases from the reaction vessel were slowly released in a 10 ml/min He flux and analyzed using mass spectrometer”

Comment 4. The durability test under flow gas mode is strongly encouraged.

Reply. *The methane conversion was below the detection limit in flow experiments. That was the reason, why the durability tests were conducted in batch experiments. No deactivation was observed after 5 consecutive cycles.*

Comment 5. The mechanism discussion neglected the by-product.

Reply: *Methane photocatalytic oxidation over the Zn-HPW/TiO₂ catalysts yields carbon dioxide in addition to CO.*

Revision: P. 20-21. *“Carbon dioxide also observed in methane photo-oxidation can produced either directly from methane or through methane consecutive oxidation to CO and then to CO₂. On TiO₂ and HPW, a significant fraction of CO₂ is probably produced via direct methane oxidation, while over Zn/HPW-TiO₂, CO₂ seems to primarily occur via CO secondary oxidation. Indeed, **Figure S5b, SM** suggests that CO₂ is mostly produced as a secondary product of methane photo-oxidation over the zinc-containing catalysts. The mechanism of the CO total oxidation to CO₂ over TiO₂ has been investigated in several previous reports [57, 58, 59]. Linsebigler [58, 59] et al. have reported that the reaction proceeds over on vacancy defect sites on the TiO₂ surface and involves O₂⁻ surface species.”*

Reviewer #3 (Remarks to the Author):

General points:

I have read a few times the Ms and the opinion about this study is stated below.

The experiments reported seem feasible and the results reported possible.

The Ms is of general interest for the readership working in the area of catalytic conversion of available materials to valuable chemical products. The English used is appropriate as well as the techniques to evaluate the reaction and the surface of the catalyst used. The authors report the selective methane oxidation at ambient conditions to obtain CO on Zn-polytungstate-titania in a reaction that seems to follow the Mars-Krevelen mechanism. This work reports the light induced conversion of CH₄ into CO with marginal CO production, and this is an interesting observation.

Reply.

We are grateful to Reviewer 3 for the scientific evaluation of our work and her/his help with our manuscript.

Specific points:

Comment 1. p1: Abstract: Delete cut the first 5 lines, they do not condense the results obtained during the course of this study.

Revision:

The abstract was revised. The introductive part was significantly reduced.

Comment 2. P.3-5: Introduction: Eliminate the prehistory of the problem at hand, cut the introduction by 40% cite items directly related to the problem at hand, beginning after reference 24.

Revision:

Introduction was shortened in line with Reviewer's suggestion. The "prehistory" of the problem was removed.

Comment 3. P.8: Figure 4 move to support material, delete the text write-up. This Figure is not important in the context of the study.

Revision: Figure 4 was moved to the Supporting Material.

Comments. p.10: Fig. 7a/7b move to support material, delete the text write-up. This Figure is also not important for the continuity of exposition of the Ms-material.

p.10-14 Discussions. Make them more pointed and precise cut text by 1/3.

p.17-17: Delete some details in the FTIR section, too many details are given. Concentrate in the main results.

p.17/19: Discussion: It is dispersed. Group 4 o 5 points of interests and condense the material cutting 1/4 of the actual section in the Ms.

p.20: Delete last paragraph of the Ms lines 400-411. Speculative and the authors know this.

Revision: *The manuscript was corrected in line with the comments and suggestions of Reviewer 3.*

REVIEWERS' COMMENTS:

Reviewer #1 (Remarks to the Author):

All the comments and questions from the reviewers have been addressed well by the authors. The revised manuscript is now clearly highlighting the important part of the findings. Several experiments were carried out to clarify the mechanisms and the active sites of the photocatalyst. The added experimental results supported the previous claims and enriched the discussion. Based on these points, I recommend the acceptance of the manuscript after the authors make some minor revisions below.

1. Page 4, line 71, omit the "practical for industrial applications" since the yield of 3-4% and quantum yield of 7.1% still not feasible for application in industry. Authors could use different term to highlight this part, such as "potential", or "attractive", etc.
2. Page 7, line 134-136, it was stated that better dispersion of Zn species is expected with higher HPW/TiO₂ ratio. However, the current result showed that the optimum ration was 0.3, and the higher ratio of 0.6 decreased the CO production. Add brief explanation about this result, so that the data and the discussion matched well.

Reviewer #2 (Remarks to the Author):

The authors have addressed most of my concerns and I would feel comfortable to see the manuscript published whilst the following problems are correctly revised:

1. Line 71: The authors claimed that "...make it practical for industrial applications." I totally cannot agree on this. What the results the authors presented are difinitely far from practical applications. For example, the low Q.E. under UV light and the lacking of durability measurement.
2. Line 208: band gas should be band gap.
3. Figure S13: lacking the label of a and b in the figure.
4. Add the curve of CH₄ in the Figures 6 and 7.

Zhiguo Yi

REPLIES TO REVIEWERS' COMMENTS:

Reviewer #1 (Remarks to the Author):

All the comments and questions from the reviewers have been addressed well by the authors. The revised manuscript is now clearly highlighting the important part of the findings. Several experiments were carried out to clarify the mechanisms and the active sites of the photocatalyst. The added experimental results supported the previous claims and enriched the discussion. Based on these points, I recommend the acceptance of the manuscript after the authors make some minor revisions below.

Comment 1. Page 4, line 71, omit the “practical for industrial applications” since the yield of 3-4% and quantum yield of 7.1% still not feasible for application in industry. Authors could use different term to highlight this part, such as “potential”, or “attractive”, etc.

Revision. The manuscript was corrected. In the corrected manuscript, we only emphasize potential future application of this reaction.

P.4. “High carbon monoxide yields (up to 3-4%), high quantum efficiency (QE=7.1% at 362 nm) and extended catalyst stability make it potentially interesting in the future for practical applications.”

Comment 2. Page 7, line 134-136, it was stated that better dispersion of Zn species is expected with higher HPW/TiO₂ ratio. However, the current result showed that the optimum ration was 0.3, and the higher ratio of 0.6 decreased the CO production. Add brief explanation about this result, so that the data and the discussion matched well.

Revision:P.5 “Some small decrease in the rate of CO production at higher HPW/TiO₂ ratio can be due to the formation of larger HPW clusters which would affect electron transfer from TiO₂ to the Zn species.”

Reviewer #2 (Remarks to the Author):

The authors have addressed most of my concerns and I would feel comfortable to see the manuscript published whilst the following problems are correctly revised:

Comment 1. Line 71: The authors claimed that "...make it practical for industrial applications." I totally cannot agree on this. What the results the authors presented are definitely far from practical applications. For example, the low Q.E. under UV light and the lacking of durability measurement.

Reply. Yes, the manuscript was corrected. See Reply to Comment 1 of Referee 1

Comment 2. Line 208: band gas should be band gap.

Revision: corrected

Comment 3. Figure S13: lacking the label of a and b in the figure.

Revision: corrected

Comment 4. Add the curve of CH₄ in the Figures 6 and 7.

Reply. We used pure methane. The variation of methane concentration was very small compared with CO and CO₂ concentrations to plot in the same Figure.